# Effects of New Compounds into Substrates on Seedling Qualities for Efficient Transplanting

Luhua Han [1,*] 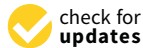, Menghan Mo [1], Yansu Gao [1], Haorui Ma [1], Daqian Xiang [1], Guoxin Ma [1] and Hanping Mao [1,2,*]

[1] Key Laboratory of Modern Agriculture Equipment and Technology, Ministry of Education, Jiangsu University, Zhenjiang 212013, China; momenghan666@gmail.com (M.M.); gyansu010526@gmail.com (Y.G.); mahaorui666@gmail.com (H.M.); godqing121@gmail.com (D.X.); 2111716008@stmail.ujs.edu.cn (G.M.)

[2] High-Tech Key Laboratory of Agricultural Equipment and Intelligence of Jiangsu Province, Zhenjiang 212013, China

\* Correspondence: hanlh@ujs.edu.cn (L.H.); maohp@ujs.edu.cn (H.M.); Tel.: +86-0511-88797338 (L.H.)

**Abstract:** Automating vegetable seedling transplanting has led to labor-saving opportunities and improved productivity. Some changes in seedling agronomy are necessary for efficient transplanting. In this study, the local nursery substrates were added with the herbaceous peat, the sphagnum peat, and the coir peat, respectively. Effects of the new compound substrates were investigated on the seedling growth qualities and the root substrate strength. In the results, we found that the addition of three compound mediums significantly affected the physiochemical properties of the original substrates. Under the same conditions of cultivating seedlings, appropriate additions of new compounds promoted the seedling growth. Moreover, deficient or excessive additions inhibited the growing development of seedlings and their roots. The corresponding additions also improved the structural characteristics of the root lumps. Compared with the two other compounds, the nursery substrates added with the sphagnum peat were optimized in contribution to the seedling qualities and the root substrate strengths. As the local substrate and the sphagnum peat were mixed at a volume ratio of 2:1, the dry matter accumulation of seedlings was 2.18 times more than the original. Their root lumps had the best consolidation strength. This new compound of substrates may be an effective application for the necessary qualities of seedlings for automatic transplanting.

**Keywords:** vegetable; seedling quality; automatic transplanting; substrate improvement; consolidation strength

## 1. Introduction

Seedling transplanting is a key technical step of vegetable production, which has the comprehensive benefits of compensating for the climate and making crops grow earlier [1,2]. Since plug seedlings with better root systems might be handled mechanically, they have been widely applied in North America, Europe, and Asia. According to the statistics, China produces up to 350 billion plants of professional plug seedlings annually [3]. Therefore, it is particularly important to ensure the timely and efficient transplanting of these seedlings. The process of seedling transplanting is to extract the seedlings from a growing plug tray and plant them into the large flower pots or the soil in the field. If the operation of vegetable transplanting on a large commercial scale is performed manually, it would be a laborious and time-consuming field operation [4]. Manual transplantations of different operating techniques are also less uniform compared to mechanical transplanters [2,5]. With the shortage of skilled labor and increasing labor costs, it is critical for China and other vegetable-producing countries to develop automatic transplanters, allowing for high-speed operations and labor-saving opportunities [1,6–8]. However, it was found that certain seedling characteristics, especially the development of root lumps (the root substrate complex shaped in the plug tray cell), significantly determined the working precision of the machine, in research on the development of automatic transplanters [9,10]. The current



automatic transplanting devices mainly handle seedlings by their roots and use needles to penetrate the root lumps for seedling extraction [2,5–12]. Therefore, some changes in seedling production are necessary if the full potential of automation in transplanting is to be realized [9,13].

Early research on the growth and physiological characteristics of plug seedlings have begun several years ago [3,14,15]. Modern high-tech means are also used to develop seedling technology. X-ray microscopic-computed tomography has been used to explore the interacting effects of soil texture and bulk density on root system development in tomato seedlings [16]. In order to adapt to the changes of the external environment, the growth and development of the root system of seedlings can be regulated at the molecular scale [17]. According to the requirements of vegetable planting, we can cultivate plug seedlings in a modern greenhouse at any time of every year. As the growth medium is loosely filled by substrate mixtures, the root system is a major component for bearing force while the seedlings are lifted [9,18]. Consequently, there is a need for a vegetable-specific study on root substrate qualities for compatibility with the transplanters [9,19,20]. Takahiro et al. studied the morphological and physical properties of various cabbage plug seedlings in different growing stages and found the fully intertwined root lumps were suitable properties for automatic transplantation [21]. Min et al. investigated the suitability of horticultural main organic substrate materials for the development of proper root lumps for working with the bulb onion transplanter [22]. It was found that the components of sphagnum moss could improve the root substrate cohesion that would give more weight of the root part during mechanical transplanting of young onion seedlings in the field. Qu et al. studied the rules of overall compressive strength from biodegradable glued substrate masses, and found that compressive strength with 50% glue was above 0.14 MPa, which theoretically met the grabbing requirements of manipulators [23]. Ma et al. explored the effects of compound biochar substrates on the root growths of cucumber plug seedlings; the compressive strengths of the substrates with 20% and 10% 'biochar-treated' was much better than others, especially that of 40% and 50% 'biochar-treated', which efficiently satisfied the requirements of automatic seedling picking [24]. With the development of high-speed transplanters, we need to further strengthen the integration between transplanters and seedling agronomy [5,19,25]. It is time to further recognize and emphasize the features of this growing system that aids in the engineering of automatic transplanting systems.

Targeting the special requirements by automatic transplanters for high-quality seedling cultivation, this study was conducted to analyze and evaluate the seedling qualities and the root substrate strengths under different compound substrate treatments. The corresponding research provides a basis for deep integration between the current seedling agronomy and automatic transplanting technology.

## 2. Materials and Methods

### 2.1. Materials and Scheme

The local nursery substrate bought from Xiangzheng Agriculture Technology Co., Ltd. (Hunan, China) was set as the control check (CK). Its composition included the peat moss, the perlite, the vermiculite, and the worm cast, which were mixed at certain volume proportions. The substrate was featured by organic matter ≥ 20%, pH 5.5–6.5, and electrical conductivity (EC) <1 mS/cm. The used compound horticultural mediums were widely used in the world, such as the herbaceous peat, the sphagnum peat, the coir peat, etc. The herbaceous peat, marked as the A-amended mix, was the rotten plant nutrient soil; it is rich in herb decomposed mass and is texturally loose. The sphagnum peat, marked as the B-amended mix, is produced by PINDSTRUP Company, Denmark. The substrate was featured by particle sizes of 0–10 mm and pH 5.5–6.0. The coir peat, marked as the C-amended mix, is produced by Galuku Pty, Ltd. (Sydney, Australia), which is featured by the air permeability of 20–30%, pH 6.0–6.6, and electrical conductivity (EC) <0.5 mS/cm.

Experiments were conducted in the intelligent Venlo glass greenhouse of Jiangsu University from September to November 2021. In preparation for treatments, each mixture

was wind-dried and the commercial substrate (CK) was used as the main material. The new compound substrate treatments of CK:A-amended mix (B-amended mix or C-amended mix) were mixed at some equal volume ratios of 3:1, 2:1, and 1:1, respectively. Finally, 10 substrate mixtures (Figure S1) were prepared for the treatments of the CK, A31 (CK:A-amended mix, 3:1), A21 (CK:A-amended mix, 2:1), A11 (CK:A-amended mix, 1:1), B31 (CK:B-amended mix, 3:1), B21 (CK:B-amended mix, 2:1), B11 (CK:B-amended mix, 1:1), C31 (CK:C-amended mix, 3:1), C21 (CK:C-amended mix, 2:1), and C11 (CK:C-amended mix, 1:1), respectively. The 128-cell trays were used with the cell dimensions of 42 mm height × 32 mm top. Each treatment was set with 1 tray and repeated 3 times. The seedling variety was the Hezuo 906 tomato. Seeds were sown into each tray cell containing 22 mL of substrates and then covered with about 2 mm of fine vermiculite. The sown plug trays were placed in the seedling beds maintained at 26 ± 2 °C for germination. Seedling growth temperatures were 24 ± 2 °C in the day and 16 ± 2 °C at night, respectively, with 65% to 75% relative humidity. Finally, the tomato plug seedlings were produced with 33 day-growth after seedings and the following 4 days of 'tempering'. Irrigated before testing, the moisture content of the root lumps was kept at a moderate range of 60 ± 2%. At the room temperature, a batch of root lumps was checked by using the dry–wet gravimetric method. As the required humidity range was reached, the corresponding test was conducted.

### 2.2. Measurement Indices and Methods

The overall technology route of this study is shown in Figure 1. According to the formulated volume ratio requirements, the local nursery substrate and the compound horticultural mediums were uniformly mixed. Thus, the new compound substrates were prepared. The physicochemical properties of each substrate treatment were analyzed, and then seedling qualities of different substrate conditions were investigated. Further, the compressive mechanical properties of seedling root lumps were strictly tested and analyzed based on the operation needs of automatic transplanting.

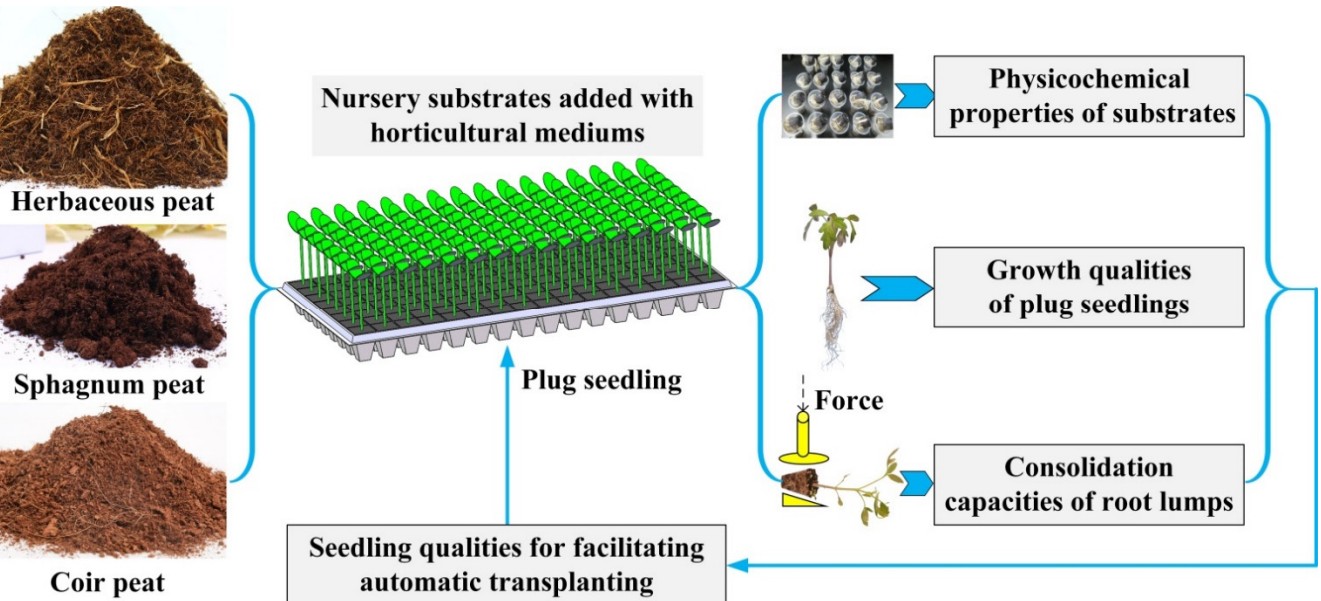

**Figure 1.** The overall technology route of studying the seedling effects of the nursery substrates added with three horticultural mediums.

### 2.2.1. Physicochemical Properties of Substrates

The substrate bulk density (BD) was computed as the naturally-dried mass per unit volume. As for air-filled porosity (AFP) and water-holding porosity (WHP), a tested sample was soaked in distilled water until saturation, and then the weight after soaking (24 h),

the weight after water dripping (24 h), and the weight after drying were measured. Total porosity (TP) was the sum of air-filled porosity (AFP) and water-holding porosity (WHP).

The chemical properties of the substrate samples were measured at room temperature (about 26.5 °C) (Figure S1). During measurements of pH and the electrical conductivity (EC), the tested sample and distilled water were mixed at a volume ratio of 1:5, and then vibrated for 30 min. After that, pH was monitored using a PH100A pH meter, and EC was measured using a CT-20 EC meter (both Shanghai Lichen Instrument Technology Co., Ltd., Shanghai, China).

### 2.2.2. Growth Qualities of Seedlings

Growth characteristics of seedlings were measured for 20 samples indoors. Stem height (SH), defined as the length from the root system to the growing point, was measured using an electronic digital display vernier caliper. Stem diameter (SD), defined as the largest stem node thickness in parallel to the cotyledon direction, was also monitored. The shoot fresh weight (SFW) and root fresh weight (RFW) were measured using an electronic balance. Shoot dry weight (SDW) and root dry weight (RDW, after being washed) were determined after as a fresh plant sample was green-removed in an oven at 105 °C for 15 min and thermostatically placed at 80 °C for 24 h. The total dry weight (TDW) was calculated by adding up the SDW and RDW. The growth index (GI) was calculated as: GI = (SD/SH + RDW/SDW)/TDW [26]. Higher growth indices indicate that the aboveground part is robust, and the underground part has well-developed roots.

### 2.2.3. Mechanical Properties of Root Lumps

Generally, the limitation of the size and shape of the plug tray cells as well as the age of the seedlings can force their roots to coil around the loose substrate particles into the composite structure [9]. During the automatic transplanting operation, the root lumps were penetrated and grasped by the pick-up device from the tray cells, which need to be stress-tolerant [5,7,10]. Here, some mechanical properties of the root lumps were characterized using the method of the texture profile analysis (TPA) being popular for determining the molded subject texture in two compression loads [27]. A total of 20 samples were tested for each treatment. The corresponding mechanical curves were obtained. Moreover, typical mechanical indices of the root substrate consolidation capacities were analyzed, which might relate to the compressive strength and texture of the root substrate composite systems. The loading velocity in testing was 0.5 mm/s (quasistatic loading/unloading) and the compression deformation of the root lumps was set at 10 mm.

### 2.3. Statistical Analyses

The experiments were set up in a completely randomized block design. Test data were recorded on EXCEL 2016 and analyzed by using SPSS (version 21, SPSS Inc., Chicago, IL, USA). The differences among these between the addition proportions of new compound substrates were studied by the Duncan multiple comparisons.

## 3. Results and Discussion

### 3.1. Physicochemical Properties of Substrate Treatments

As shown in Table 1, addition with three horticultural mediums significantly affected the physicochemical properties of the original substrates. These new compound mediums changed the basic particle properties of the original substrates in different ways.

Generally, as the dosage of the nursery substrates added with horticultural mediums rose, the bulk density (BD) of A-amended mixes slightly increased, and those of the substrate mixtures with addictive B-amended mixes or C-amended mixes gradually decreased. An ideal substrate for raising seedlings must have a bulk density (BD) of 0.1–0.8 g/cm$^3$ [16,24], which can be met by all treatments in our study. When the new compound treatment was set at A11, the bulk density (BD) of the substrate achieved the maximum of 0.2239 g/cm$^3$. It exceeded that of the CK. When the substrate treatment of

C-amended mixes was C11, the bulk density (BD) of the substrate mixture minimized to 0.1585 g/cm$^3$, which was far lower than that of the CK. So the nursery substrates of A-amended mixes and C-amended mixes oppositely affected their bulk densities. The reason may be that A-amended mixes (plant fiber soil) contained some heavy soil grains, which undoubtedly increased the weight of the substrate mixture. In comparison, C-amended mixes were those light components of coir nuts, which were fragmental and also resilient shredded. Hence, the addition with C-amended mixes made the substrate texture loose. The nursery substrates of B-amended mixes did not have many changes of the bulk density (BD).

**Table 1.** Physicochemical properties of substrates treated with different additions.

| Treatments | BD, g/cm$^3$ | AFP, % | WHP, % | TP, % | pH | EC, mS/cm |
|---|---|---|---|---|---|---|
| CK | 0.2106 b | 21.92 a | 39.74 e | 61.65 c | 6.44 bcd | 0.37 bc |
| A31 | 0.2044 c | 14.32 d | 40.73 de | 55.05 d | 6.41 cd | 0.32 e |
| A21 | 0.2052 c | 18.56 bc | 42.70 cde | 61.26 c | 6.62 b | 0.31 e |
| A11 | 0.2239 a | 19.96 ab | 47.28 bc | 67.24 b | 6.94 a | 0.22 f |
| B31 | 0.2096 b | 16.41 cd | 44.53 bcd | 60.94 c | 6.18 ef | 0.33 de |
| B21 | 0.2024 c | 17.17 bc | 45.30 bcd | 62.47 c | 6.04 fg | 0.40 b |
| B11 | 0.1937 e | 18.31 bc | 49.18 b | 67.49 b | 5.94 g | 0.53 a |
| C31 | 0.1986 d | 21.67 a | 44.75 bcd | 66.41 b | 6.50 bc | 0.31 e |
| C21 | 0.1858 f | 19.75 ab | 48.55 b | 68.30 b | 6.45 bcd | 0.40 b |
| C11 | 0.1585 g | 18.13 bc | 56.21 a | 74.34 a | 6.27 de | 0.52 a |

Note: BD: bulk density; AFP: air-filled porosity; WHP: water-holding porosity; TP: total porosity; EC: electrical conductivity. The data in the table are the average of five samples of the same factors. The same letters indicate no significant difference at $p < 0.05$ level along the columns by Duncan's multiple comparison method.

In all of the substrate treatments, the air-filled porosity (AFP) of the CK achieved the maximum of 21.92%. This was because the commercial substrate was composed of irregular loose particles. The scanning electron microscopy (SEM) (Figure S2) showed the loose morphology of the CK in Figure 2a. These particles accumulated into disorderly layers, which moderately increased the air-filled porosity (AFP). The water-holding ability was the weakest in the CK. As the adding proportion increased, the water-holding ability of the commercial substrates with each amended mix was significantly improved. SEM (Figure 2b) showed that A-amended mixes contained the decomposed parts from dead vascular bundle plants (e.g., sedges, reeds) and, thus, had air-filled vascular bundles. B-amended mixes resulted from the decomposition of dead moss plants, which reserved high free porosity and were texturally loose (Figure 2c). Moreover, the thin-walled cellular pores can well store and transport water. C-amended mixes were of fragmented structure, which could messily accumulate much water-holding pores (Figure 2d). Hence, these three types of horticultural biomaterials with irregular morphology and free porosity can largely improve the water holding ability of the mixed substrate, which would be favorable for water and fertilizer management during seedling growth [14,15,19]. The contribution to the water holding ability of the substrate ranked as C-amended mixes > B-amended mixes > A-amended mixes. When the substrate treatment of C-amended mixes was C11, the water holding ability maximized to 56.21%, which was 1.41 times that of the CK. The total porosity (TP) changed in similar rules, liking the air-filled porosity (AFP). Overall, the studied total porosity of each substrate mixture was qualified for seedling growth [19].

In terms of chemical properties, the pH of A-amended mixes gradually increased as the addition ratios rose, and then those of B-amended mixes or C-amended mixes decreased. When the substrate treatments of A-amended mixes were A11 and A21, their pH values were up to 6.94 and 6.62, respectively. However, previous studies recommended pH for normal growth of tomato seedlings to keep at 5.5–6.5 [19]. Apparently, these two substrate treatments of A11 and A21were slightly unqualified. Nevertheless, the agricultural production components should be modified to adjust their acidity and basicity to meet the seedling raising requirements before use [19,28]. In comparison, the pH values of B-amended mixes or C-amended mixes were steadily moderate from 5.94 to 6.50, which

could meet the raising requirements of tomato seedlings. Moreover, with the increase of the addition ratio, the EC values of A-amended mixes gradually decreased. When the ratio of substrates added with A-amended mixes was 1:1, the EC minimized to 0.22 mS/cm. The possible reason may be that A-amended mixes contains some barren soil grains, to inhibit the soluble ionic concentrations [24,28]. So the nutrients in this case were insufficient. As the ratios of the addictive rose, the EC values of B-amended or C-amended mixes gradually increased. When the new compound treatment was set at B11 and C11, their EC values were 0.53 and 0.52 mS/cm, respectively, which were far higher than that of the CK. Admittedly, the incorporation of B-amended mixes and C-amended mixes into the commercial substrates could improve the soluble ionic concentration, which would not exceed the range of EC in normal seedling growth (EC below 2.5 mS/cm) [28].

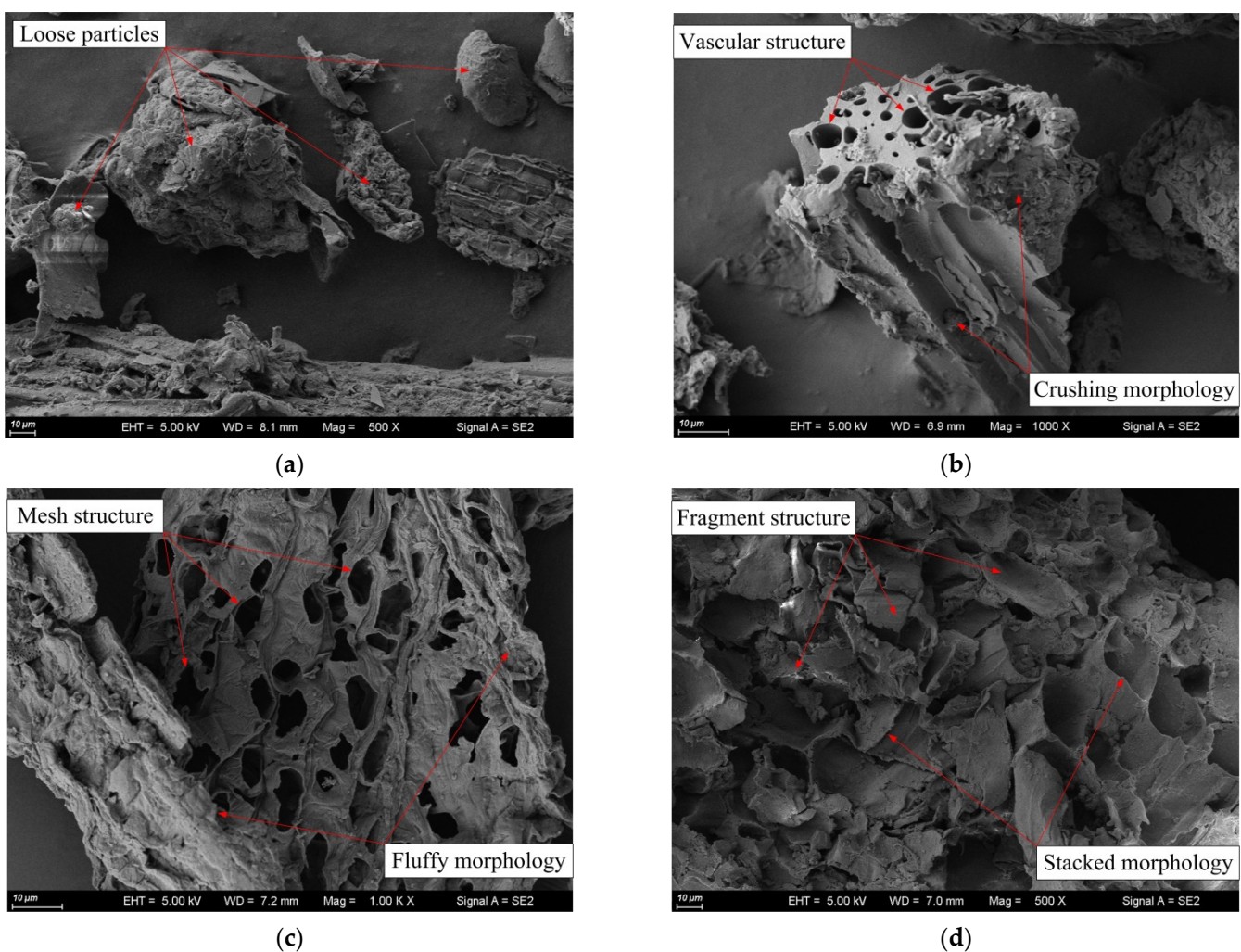

**Figure 2.** SEM images of the major materials mixed into substrates: (**a**) local substrate; (**b**) herbaceous peat; (**c**) sphagnum peat; (**d**) coir peat.

### 3.2. Growth Qualities of Plug Seedlings under Compound Substrate Treatments

Under the same production conditions (Figure S3), the plant and root growth of plug seedlings cultivated by different substrate treatments both differed significantly (Table 2).

As the dosage of the nursery substrates added with horticultural mediums rose, the stem diameter (SD) of seedlings grown in the A-amended or B-amended mixes first increased and then decreased, but that of the C-amended mixes gradually declined. For the stem height (SH), the effects of the three amended mixtures showed different changes. The addition of horticultural mediums could moderately increase seedling stem height (SH). When the ratio of substrates added with B-amended mixes was 2:1, the stem diameter



(SD) and stem height (SH) of seedlings both maximized, which were 2.71 and 105.88 mm, respectively. They increased by 1.35 and 1.37 times in comparison with the CK. As the new mixing treatment was set at the C21, the seedling growth was the weakest than other compound substrates. Overall, the nursery substrates added with B-amended mixes were suitable for the growth development of tomato seedlings. The main underlying reason for such a large increase was that B-amended mixes could always keep at a high free porosity, which would well hold water and provide nutrients throughout the seedling growth period [14,24]. However, the seedling qualities were relatively weak when the C-amended mixes were excessive at C11. This might be because the compound substrates of the C-amended mixes held much water, making flooding stress. As a result, the root system of tomato seedlings cannot well adapt to the flooding environment, which thereby hinders normal growth and development [29].

**Table 2.** Physical properties of plug seedlings treated with different additions.

| Treatment | SD, mm | SH, mm | SFW, g | RFW, g | SDW, g | RDW, g | TDW, g |
|---|---|---|---|---|---|---|---|
| CK | 2.07 c | 77.44 cd | 0.41 def | 0.20 d | 0.0246 e | 0.0176 cd | 0.0422 e |
| A31 | 1.92 cd | 78.05 cd | 0.47 d | 0.23 d | 0.0302 de | 0.0184 bcd | 0.0486 de |
| A21 | 2.42 b | 96.63 b | 0.68 c | 0.32 bc | 0.0482 c | 0.0212 b | 0.0695 c |
| A11 | 2.36 b | 98.37 b | 0.70 c | 0.30 c | 0.0512 c | 0.0188 bcd | 0.0700 c |
| B31 | 2.01 cd | 82.36 c | 0.46 de | 0.30 c | 0.0269 de | 0.0164 d | 0.0433 e |
| B21 | 2.71 a | 105.88 a | 0.90 a | 0.39 a | 0.0670 a | 0.0248 a | 0.0918 a |
| B11 | 2.69 a | 103.35 ab | 0.80 b | 0.35 b | 0.0610 b | 0.0215 b | 0.0825 b |
| C31 | 2.08 c | 70.30 e | 0.39 ef | 0.22 d | 0.0276 de | 0.0164 d | 0.0439 e |
| C21 | 1.95 cd | 73.75 de | 0.42 def | 0.24 d | 0.0328 d | 0.0206 bc | 0.0534 d |
| C11 | 1.86 d | 69.31 e | 0.38 f | 0.21 d | 0.0284 de | 0.0188 bcd | 0.0472 de |

Note: SD: stem diameter; SH: stem height; SFW: shoot fresh weight; RFW: root fresh weight; SDW: shoot dry weight; RDW: root dry weight; TDW: total dry weight. The data in the table are the average of 20 samples of the same factors. The same letters indicate no significant difference at $p < 0.05$ level along the columns by Duncan's multiple comparison method.

The appropriate addition of auxiliary substrate materials was beneficial to the dry matter accumulation of tomato seedlings. The shoot fresh weight (SFW) and root fresh weight (RFW) of the seedlings grown in the compound substrates of A-amended mixes or B-amended mixes were all larger than those in the CK. Contrarily, the addition of C-amended mixes did not significantly affect the dry matter accumulation of seedling plants in comparison with the CK. As the mixing ratio rose, the shoot dry weight (SDW) and root dry weight (RDW) of seedlings grown in each compound substrate treatment all first increased and then decreased. When the ratio of substrates added with B-amended mixes was 2:1, the sum of total dry matters in the tomato seedlings was 0.0918 g, which was 2.18 times than that of the CK. Moreover, this situation was far larger in comparison with the treatments added with addictive A-amended mixes or **C**-amended mixes. These two substrate treatments of B11 and C21 did not significantly affect the sum of dry matters in comparison with the CK. The effects under all other ratios were superior to the CK. Generally, the accumulative dry matters of plug seedlings grown on the compound substrates could be larger than those of the CK. Moreover, seedling qualities in the compound substrates of B-amended mixes were higher than other two mixes. The possible reason is that B-amended mixes have high permeability, stable pH, and a long period of wall-breaking cell support, which promote seedling growth [24,28].

Figure 3a compares the growing morphology of tomato seedlings grown under different substrate treatments. Clearly, the appropriate addition of auxiliary substrate materials, such as A21 and B21, could cultivate sturdy seedlings growing in tray cells and develop the large root volume [19]. However, little or excessive addition of auxiliary substrate materials, such as A11, B31, C31, and C11, did not reach such good effects. The seedling index was considered to objectively reflect the growing qualities of seedlings. On the basis of the measured growing traits, the corresponding seedling indices were calculated [26]. The addition of auxiliary substrate materials significantly affected the seedling index (Figure 3b). The

seedling index was optimized in B21 containing sphagnum peat (mean 0.0371), followed by the C21 with C-amended mixes (mean 0.0357). They were both higher than the CK and other mixing treatments. Compared with the CK, the seedling index was not significantly different between treatments A31 and A21 (both added with plant fiber soil), and also treatment B11 (added with sphagnum peat) or treatment C11 (added with coir peat). Viewed from the strong seedling cultivation, the substrate treatments of A11 (added with plant fiber soil), B31 (added with sphagnum peat), and C31 (added with coir peat) all produced slightly thin and weak plants. In these cases, the seedling index was lower than that of the CK. So they were not suitable to be used as the substrate modifications to improve seedling quality.

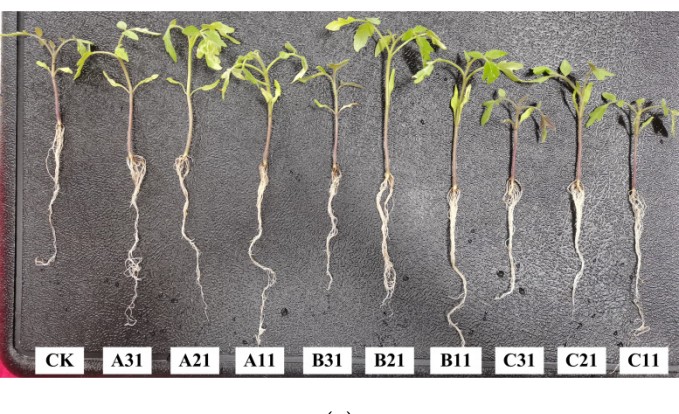

(**a**)

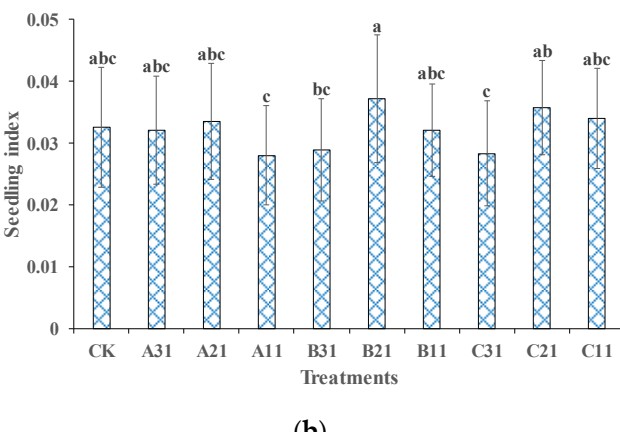

(**b**)

**Figure 3.** Growth characteristics of tomato seedlings raised in different substrates: (**a**) growth morphology of tomato seedling; (**b**) seedling index: the same letters indicate no significant difference at *p* < 0.05 level by Duncan's multiple comparison method.

### 3.3. Consolidation Capacities of Root Lumps under Compound Substrate Treatments

Figure 4 shows the force–time curves of secondary compression tests for root lumps of tomato seedlings under different substrate treatments (Figure S4). In the compression loading process, the anti-pressure ability of root lumps varied uniformly. With the increase of the compression deformation, their anti-pressure abilities were significantly strengthened. There were no obvious yield failure points in the whole compression loading [30]. After two loading tests, the resistance capacity with deformation gradually increased, which showed certain biological compaction and hardening. For the root system in the tray cells, the slide, collapse, and rearrangement of substrate particles may be the main reason for the softening phenomenon to the hardening. The resistance capacity with deformation increased slowly at first, showing certain characteristics of biological yield softening; and increased significantly at last, which showed certain biological compaction and hardening [19]. Thus it could be seen that root lumps liking other organisms have certain flexibility [9].

According to the force–time curves of secondary compression tests, the consolidation capacities of root lumps were analyzed and calculated. The compressive hardness of root lumps was measured from the peak force. The addition of auxiliary substrate materials significantly affected the hardness of the root substrate composite structures (Figure 5a). With the increase of the addition ratio, the root substrate hardness of plug seedlings of A-amended mixes or **B**-amended mixes first increased and then decreased. However, the anti-pressure ability of C-amended mixes gradually declined. This compressive hardness of the root lumps might be consistent with their growing traits. When the substrates added with A-amended mixes and **B**-amended mixes were set at the ratios of 2:1 and 1:1, respectively, the maximum compressive hardness of the root lumps was gained. The first and second hardness processed by A21 were 19.78 N and 15.16 N, respectively. In this case, it was almost twice as hard as the CK. The possible reason was that A-amended mixes included the plant nutrient soil, making shaped root lumps difficult to compress. On

the contrary, the root lumps cultivated with the presence of C-amended mixes generally reacted to low hardness in the loading. The possible reason was that C-amended mixes of coir peats were fragmental and made to easily collapse upon compression. Moreover, the plug seedlings cultivated with the addition of C-amended mixes were moderately growing and had no good twisting roots. Thus, they could not well restrict the substrate particles against compression deformation [19,24,31].

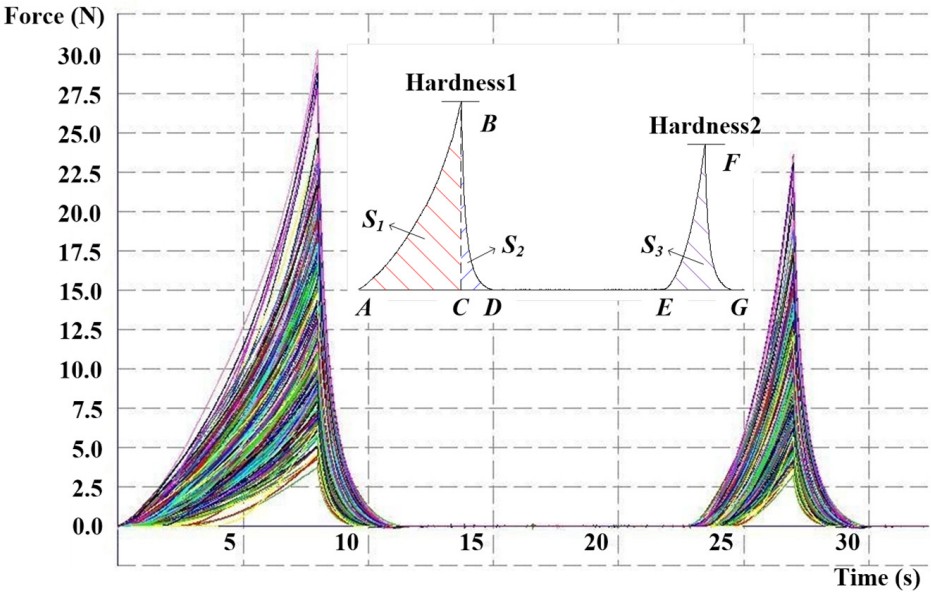

**Figure 4.** Force–time curve of secondary compression tests for root lumps of tomato seedlings raised in different substrates: line *A-B-D* is the first loading and unloading force–time curve; line *E-F-G* is the second loading and unloading force–time curve; $S_1$ is the area bounded by the force–time curve of line *A-B-C*; N.s; $S_2$ is the area bounded by the force–time curve of line *C-B-D*, N.s; $S_3$ is the area bounded by the force–time curve of line *E-F-G*, N.s; the cohesiveness is the ratio of the energy done by the second compression to the first, which is calculated by $S_3/(S_1 + S_2)$; The resilience is the ratio of the energy done by the first unloading compression to the loading, which is calculated by $S_2/S_1$.

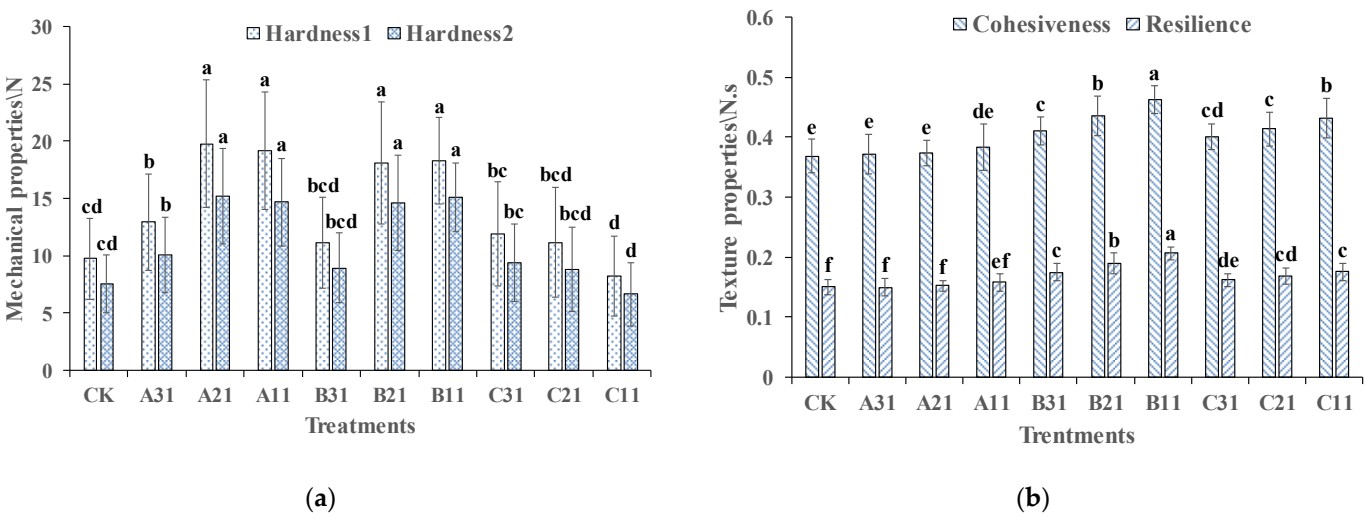

|  |  |
|:---:|:---:|
| (a) | (b) |

**Figure 5.** Analysis of consolidation capacities of root lumps raised in different substrate treatments: (**a**) mechanical properties in secondary compression tests; (**b**) texture properties in secondary compression tests.

As shown in Figure 5b, the root lumps of **B**-amended mixes had excellent cohesiveness and resilience, which were compared with other treatments. As the addition of **B**-amended

mixes increased, it was strong for the root substrate structure stability of plug seedlings. When the ratio of the commercial substrates added with the B-amended mix was 1:1, the average cohesiveness and resilience were up to 0.46 N.s and 0.21 N.s, respectively. In terms of these textures, the addition of **B**-amended mixes was 1.26 and 1.37 times higher than the CK. Although the high-quality seedlings were not produced by B11, the excellent performance of their root lumps in compression suggested that cohesion of the substrate components was important in contributing to the structural stability. The texture properties of the root lumps added with C-amended mixes were second only to that of **B**-amended mixes. Moreover, the tray root lumps cultured by these treatments all showed similar resilience and deformation rules. However, the root lumps cultivated under the substrate added with A-amended mixes were not significantly different from the CK. These root lumps under external compression showed inferior cohesiveness and resilience, which were of much biological compaction [30]. The contributions of the structural stability of the root substrate bodies ranked as B-amended mixes > C-amended mixes > A-amended mixes.

A reported greenhouse robotic mechanism was used to accomplish automatic transplanting works from the growing trays to the destination trays (Figure S5) [10]. Then the transplanted seedlings were released from 100 cm high via free dropping to 304 stainless steel plates of 2 mm thickness. The automatic transplanting and dropping fragmentation of various root lumps are showed in Figure 6. Most of the seedlings after the addition of **B**-amended mixes were basically complete. Even under the actions of transplanting and dropping impact, the root substrate structure kept its integrity [2,19]. The seedlings transplanted in this way can rapidly return to greenness after planting. It can be concluded that the percentage of successful transplanting was largely dependent on the root/growth medium portion of the seedling. The root lumps of **A**-amended mixes and **C**-amended mixes presented similar mechanical damage in comparison with the CK. The reason for these phenomena was that the cohesive strength between the substrate grains and the cell root volume was insufficient [2,10]. In order to adapt to the characteristics of objects, the flexible pick-up gripper of variable parameters should be adopted for transplanting those special root lumps with different structural strengths [32].

Based on the above studies, we could see that the compound substrates of A-amended mixes were too alkaline for seedling cultivation. It must be modified according to different seedling requirements. To facilitate the softening and reanimation after picking, the addition proportion of A-amended mixes (herbaceous peat) should be appropriately increased. The root systems of plug seedlings were often found to be distributed differently within the cell for the various substrate treatments [9,14]. As the commercial substrates were befittingly added with B-amended mixes, plug seedlings were always sturdy with rich roots, and their root lumps were featured by hardness as well as strong cohesiveness and resilience upon compression loading. It was suggested that B-amended mixes (sphagnum peat) should be mixed into the nursery substrates for a very high level of seedling quality and uniformity. Further, appropriate addition of C-amended mixes (coir peat) did improve the growth qualities of plug seedlings. However, the root lumps under C-amended mixes was of poor hardness on the whole and also did not well resist the compression deformation. As for renewable resources, C-amended mixes (coir peat) would be used together with other natural organic medium. It may be an effective way of reclamation in sustainable agriculture.

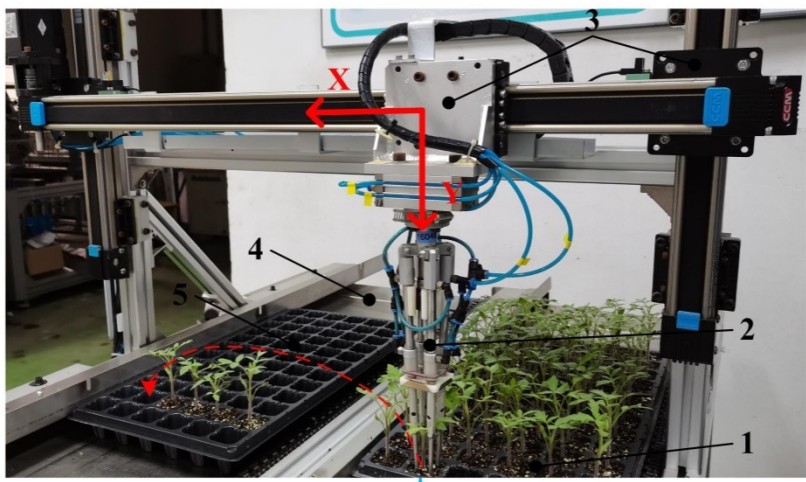

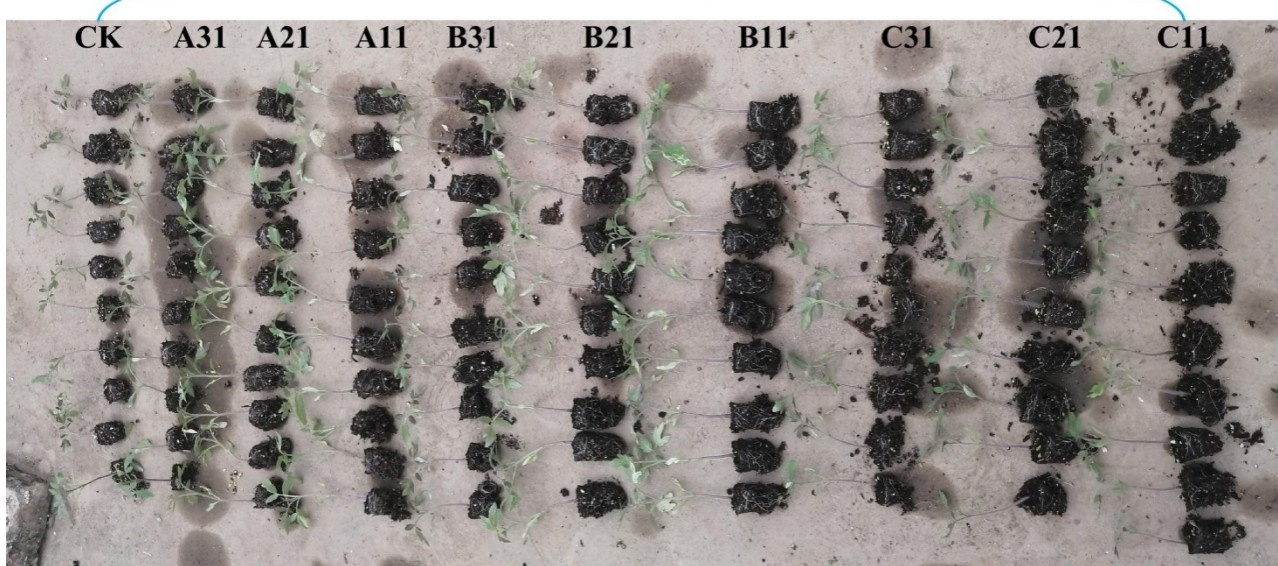

**Figure 6.** Failure morphology of various root lumps of tomato seedlings in automatic transplanting: (1) source tray; (2) seedling pick-up gripper; (3) Cartesian manipulator; (4) conveyor for seedling tray; (5) destination tray.

## 4. Conclusions

Three horticultural mediums were used to improve the nursery substrates for an extensive adaptability study on seedling qualities, for compatibility with automatic transplanting. We found that the addition of three horticultural mediums with unique morphological characteristics significantly affected the physicochemical properties of the original substrate. A variety of compound organic substrates were obtained, which were of different porosities. According to the different growth needs, the physicochemical properties of the mixed matrix should be properly regulated to strengthen the seedlings. When the ratio of the nursery substrates added to the sphagnum peat or the coir peat was 2:1, the seedling index was optimized in comparison with the control check and other mixing treatments. The substrate treatments of A11 (added with plant fiber soil), B31 (added with sphagnum peat), and C31 (added with coir peat) all produced slightly thin and weak plants. It can be concluded that the appropriate addition of auxiliary substrate materials cultivates sturdy seedling growing in tray cells. Basically, the seedlings with good growth have characteristics of consolidation and integration at their root lumps. When the commercial substrates were (befittingly) added to the sphagnum peat, plug seedlings were sturdy with rich roots, and their root lumps were featured by hardness, strong cohesiveness, and resilience upon compression

loading. The new compound substrate treatment with special seedling qualities may be easy for automatic transplanting.

**Supplementary Materials:** The following supporting information can be downloaded at: https://www.mdpi.com/article/10.3390/agronomy12050983/s1. Figure S1: Compound substrate treatments and testing for their physicochemical properties, Figure S2: SEM images of the major materials of compound substrates, Figure S3: Tomato seedlings raised in different compound substrates, Figure S4: Testing for the growth and mechanical properties of plug seedlings, Figure S5: Automatic transplanting works from the growing trays to the destination trays.

**Author Contributions:** Conceptualization, L.H. and H.M. (Hanping Mao); data curation, M.M. and H.M. (Haorui Ma); formal analysis, L.H. and M.M.; funding acquisition, L.H.; investigation, Y.G. and G.M.; methodology, M.M., Y.G. and H.M. (Haorui Ma); supervision, H.M. (Hanping Mao); project administration, L.H. and H.M. (Hanping Mao); writing—original draft preparation, L.H., M.M. and D.X.; writing—review and editing, Y.G. and H.M. (Haorui Ma). All authors have read and agreed to the published version of the manuscript.

**Funding:** This work was funded by the National Natural Science Foundation of China (no. 51975258), the Jiangsu Demonstration Project of Modern Agricultural Machinery Equipment and Technology (no. NJ2019-19), and the Open Fund of High-tech Key Laboratory of Agricultural Equipment and Intelligentization of Jiangsu Province (no. JNZ201910).

**Acknowledgments:** The authors would like to thank the Key Laboratory of Agricultural Engineering in Jiangsu University for supporting the experimental conditions of this research.

**Conflicts of Interest:** The authors declare no conflict of interest.

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
