# Peer review of "Effects of New Compounds into Substrates on Seedling Qualities for Efficient Transplanting"

_agronomy, doi:10.3390/agronomy12050983_

Round 1

Reviewer 1 Report

no fruther comments

paper is well done

Author Response

Dear reviewer,

Thank you for your the recognition of our manuscript.

According to the reviewing comment, we only check english language and style of our manuscript.

Thank you again for your positive comments and valuable suggestions to improve the quality of our manuscript.

Reviewer 2 Report

Dear Authors,

The presented manuscript is devoted to investigate the effects of the addition of new compounds to substrate and the influence of the substrate composition on the seedling quality and the root substrate strength. Improving the nursery substrates is one of the steps to seedling technology development. The study presents new knowledge in this field.

The manuscript has good quality; it needs only several minor corrections.

I recommend shortening the title a bit, it is not necessary to have everything in the name.

The abstract requires some smoothing of the wording, the text is sometimes rough. For example: horticultural mediums, „Under the same? nursery conditions, some? appropriate? additions could? promote“ –  present results, not such vague statements, as well „And the deficient or excessive additions were to inhibit...“, „Generally?, the nursery substrates added with the sphagnum peat were relatively? optimized“

L21, 25 and elsewhere - many readers will not understand what is meant by "root lumps", it is necessary to explain this more precisely in the methodology. Similarly, L25 "consolidation strength" - use terms known to the wider readership not only to specialists or explain.

L23 "commercial substrate" what is it?, not mentioned in the previous text. The sentence “Espetially… …. “ is too long and incomprehensible.

The Introduction is brief and clear. Materials and methods section is clearly elaborated in terms of content, structure and style. The interpretation of obtained data is logical and clear. There are only a few ambiguities and inaccuracies in the text.

L33 Unclear

L95 Unclear wording „the rotten plant nutrient soil“

L.105 How can the composition of the mixture be standardized on the basis of "volume ratios", which depends on the current state of the material, why did you not use the appropriate weight ratios, and what was the dry matter of the materials?

L122 "the compound substrates were uniformly prepared".

L118-119 How was humidity measured to keep such a narrow range? Is it gravimetric or volumetric humidity?

L131 Correct perunit

How did you standardize the volume weight of substrates, it may depend, e.g. on size and compression of original package of a substrate

L.149-150 We know from our own experience that separating roots from fibrous peat is almost impossible, wasn't it a source of error?

L153 Correct RWD, TWD

The meaning of the index could be explained (higher index=thicker, shorter, sturdy, robust stem and greater proportion of roots), also see L287 – what means „objectively“

L168 A little unusual – „analyzed on? EXCEL“, „The results were sent? to analysis“

L183 and elsewhere, I'm not a native speaker, but using the word "maximize, minimize" doesn't sound very good, does it mean the same thing as "achieve" or "drop to"?

Table 1, 2.

L.242 Not “growth”, but rather physical properties

Add explanations of property shortcuts, align numbers in columns

In some cases, I recommend reducing the number of digits after the decimal point, the readability of the tables will improve; what is the practical significance of differences in tens of thousands of g in BD, or hundredths in WHP or TP. Similarly in Tab.2 SH

Tab 2 It would be clearer to indicate the weight in mg

L258 What nutrients were added and in what amount? Or there was a different content of available nutrients in the substrates? Also L232, L338, L407

L261 But soil moisture was kept at the same level in all variants?

L266-275 Some letters are in bold, others not – unify

L272 Ground x aboveground (shoot)                                                                                                                                                                                                                                                                                                                                                                                                                                                                                                                                                     

L316 Unclear

Fig. 6 The caption describes another picture?

The conclusion summarizes the findings of this study. The addition of three horticultural media with the unique morphological characteristics significantly affected the physico-chemical properties of the original substrate. Changing the composition of the growing medium can change its properties and improve the conditions for the growth of seedlings and their automatic transplanting.

Please note, that some of the Reviewer´s comment are only suggestions and refer to the formal or technical aspects of the paper.

Author Response

Dear reviewer,

Thank you for your the recognition of our manuscript.

According to the reviewing comment, we have tried best to modify our manuscript. Please you check it again.

<Comment 1> recommend shortening the title a bit, it is not necessary to have everything in the name.

Response: According to the reviewing comment, the title has been shortened. It may focus on how to have effects on seedling qualities for efficient transplanting by adding new compounds into nursery substrates. Here, seedling qualities involve the seedling growth qualities and the root-substrate strength.

<Comment 2> The abstract requires some smoothing of the wording, the text is sometimes rough. For example: horticultural mediums, „Under the same? nursery conditions, some? appropriate? additions could? promote“ –  present results, not such vague statements, as well „And the deficient or excessive additions were to inhibit...“, „Generally?, the nursery substrates added with the sphagnum peat were relatively? optimized“

Response: According to the reviewing comment, the abstract has been revised by using the explicit text. There are many beneficial effects by comparing a variety of substrate treatments. Due to the limited number of words, the overall summary description can only be expressed in the abstract.

<Comment 3> L21, 25 and elsewhere - many readers will not understand what is meant by "root lumps", it is necessary to explain this more precisely in the methodology. Similarly, L25 "consolidation strength" - use terms known to the wider readership not only to specialists or explain.

Response: According to the reviewing comment, the phrases of root lumps and consolidation strength have explained in place.

<Comment 4> L23 "commercial substrate" what is it?, not mentioned in the previous text. The sentence “Espetially… …. “ is too long and incomprehensible.

Response: According to the reviewing comment, the commercial substrate has been changed to the local substrate in accordance with the above. The corresponding sentence has been simplified.

<Comment 5> L33 Unclear

Response: According to the reviewing comment, the corresponding sentence has been revised to clarify meaning. Is it appropriate for such modification? Please further review it.

<Comment 6> L95 Unclear wording „the rotten plant nutrient soil“

Response: According to the reviewing comment, the corresponding sentence has been revised to clarify meaning. The rotten plant nutrient soil is thoroughly decomposed under natural conditions. Is it appropriate for such modification? Please further review it.

<Comment 7> L.105 How can the composition of the mixture be standardized on the basis of "volume ratios", which depends on the current state of the material, why did you not use the appropriate weight ratios, and what was the dry matter of the materials?

Response: The substrate materials are relatively light. In gardening, the used substrates are generally mixed in terms of volume ratios. In preparation for treatments, each mixture has been wind-dried. The new compound substrate treatments are mixed at some equal volume ratios of 3:1, 2:1 and 1:1, respectively. The dry matter of the materials is mixed in a free state, which is more accurate. Some beneficial effects are studied by comparing these substrate treatments.

<Comment 8> L122 "the compound substrates were uniformly prepared".

Response: According to the reviewing comment, the corresponding sentence has been revised to clarify meaning. Is it appropriate for such modification? Please further review it.

<Comment 9> L118-119 How was humidity measured to keep such a narrow range? Is it gravimetric or volumetric humidity?

Response: According to the reviewing comment, the corresponding sentence has been revised to clarify meaning. At the room temperature, a batch of root lumps was checked by using the dry-wet gravimetric method. As the required humidity range was reached, the corresponding test was conducted. This can be done in practice.

<Comment 10> L131 Correct perunit

How did you standardize the volume weight of substrates, it may depend, e.g. on size and compression of original package of a substrate

Response: According to the reviewing comment, the corresponding phrase has been corrected. It is a very conventional method to measure the volume weight of substrates. After the substrates prepared with different proportions were configured, samples were taken immediately with the ring cutter, and we weighed the ring cutter filled with the substrate as soon as possible. At the same time, the aluminum box was used for sampling in the same substrate to determine the mass moisture content, so we can easily get the bulk density.

<Comment 11> L.149-150 We know from our own experience that separating roots from fibrous peat is almost impossible, wasn't it a source of error?

Response: In agronomy, water washing is the easiest way to separate roots. As all samples are treated by using the same operation method, the root system of seedlings can be studied by means of average analysis.

<Comment 12> L153 Correct RWD, TWD

The meaning of the index could be explained (higher index=thicker, shorter, sturdy, robust stem and greater proportion of roots), also see L287 – what means „objectively“

Response: According to the reviewing comment, the corresponding phrase has been corrected. The seedling index is considered to objectively reflect the growing qualities of seedlings. Higher growth indexes indicates that the aboveground part is robust, and the underground part has well-developed roots. The corresponding sentence has been revised to clarify meaning.

<Comment 13> L168 A little unusual – „analyzed on? EXCEL“, „The results were sent? to analysis“

Response: According to the reviewing comment, the corresponding sentence has been revised to clarify meaning. Test data were recorded on EXCEL 2016 and analyzed by using SPSS (Version 21, SPSS Inc., Chicago, USA). The differences among these between the addition proportion of new compound substrates were studied by the Duncan multiple comparisons. Is it appropriate for such modification? Please further review it.

<Comment 14> L183 and elsewhere, I'm not a native speaker, but using the word "maximize, minimize" doesn't sound very good, does it mean the same thing as "achieve" or "drop to"?

Response: According to the reviewing comment, the corresponding phrase has been corrected in line183 and elsewhere.

<Comment 15> Table 1, 2. L.242 Not “growth”, but rather physical properties

Response: According to the reviewing comment, the corresponding phrase of growth qualities has been corrected.

<Comment 16> Add explanations of property shortcuts, align numbers in columns

In some cases, I recommend reducing the number of digits after the decimal point, the readability of the tables will improve; what is the practical significance of differences in tens of thousands of g in BD, or hundredths in WHP or TP. Similarly in Tab.2 SH

Tab 2 It would be clearer to indicate the weight in mg

Response: We thank the reviewer for this suggestion. For the sake of consistency in the context, we consider using the same grams.

<Comment 17> L258 What nutrients were added and in what amount? Or there was a different content of available nutrients in the substrates? Also L232, L338, L407

Response: We thank the reviewer for this suggestion. All treatments in the experiment did not use additional fertilizers, and the nutrients mentioned in the paper were from the mixed matrix itself. Contents of available nutrients in the substrates were reflected by the EC value, the plant growth and so on. Here, it was expressed that the substrate treatments of some mixes might improve the original nutrient levels. Thus this compound substrate could promote seedling growth.

<Comment 18> L261 But soil moisture was kept at the same level in all variants?

Response: In seedling management, we found that the compound substrates of the C-amended mixes easily hold much water. It is because the C-amended mixes have strong water retention ability. So it makes flooding stress to the root system of tomato seedlings. As a result, the root system of tomato seedlings cannot well adapt to the flooding environment, which thereby hinders normal growth and development.

<Comment 19> L266-275 Some letters are in bold, others not – unify

Response: According to the reviewing comment, the corresponding letters have been corrected.

<Comment 20> L272 Ground x aboveground (shoot)

Response: According to the reviewing comment, the corresponding phrase has been corrected.

<Comment 21> L316 Unclear

Response: According to the reviewing comment, the corresponding sentence has been revised to clarify meaning. Generally, the limitation of the size and shape of the plug tray cells as well as the age of the seedlings can force their roots to coil around the loose substrate particles into the composite structure. There were no obvious yield failure points in the whole compression loading. The resistance capacity with deformation increased slowly at first, showing certain characteristics of biological yield softening; and increased significantly at last, which showed certain biological compaction and hardening. Thus it could be seen that root lumps liking other organisms have certain flexibility.

<Comment 22> Fig. 6 The caption describes another picture?

Response: We thank the reviewer for this suggestion. According to the reviewing comment, the corresponding picture in Figure 6 has been corrected.

Thank you again for your positive comments and valuable suggestions to improve the quality of our manuscript.

Dr. Han.